# INDUCTIVE BIASES FOR RELATIONAL TASKS

**Giancarlo Kerg**[1] * **Sarthak Mittal** [1] **David Rolnick** [2,4] **Yoshua Bengio** [1,3,4]
**Blake Richards** [2,4] **Guillaume Lajoie** [1,4]
Mila, Quebec AI Institute

## ABSTRACT

Current deep learning approaches have shown good in-distribution performance but struggle in out-of-distribution settings. This is especially true in the case of tasks involving abstract relations like recognizing rules in sequences, as required in many intelligence tests. In contrast, our brains are remarkably flexible at such tasks, an attribute that is likely linked to anatomical constraints on computations. Inspired by this, recent work has explored how enforcing that relational representations remain distinct from sensory representations can help artificial systems. Building on this work, we further explore and formalize the advantages afforded by "partitioned" representations of relations and sensory details. We investigate inductive biases that ensure abstract relations are learned and represented distinctly from sensory data across several neural network architectures and show that they outperform existing architectures on out-of-distribution generalization for various relational tasks. These results show that partitioning relational representations from other information streams may be a simple way to augment existing network architectures' robustness when performing relational computations.

## 1 INTRODUCTION

Human cognition is heavily dependent on the ability to understand relationships between different entities in the world, regardless of their sensory attributes (Kriete et al., 2013). This ability allows us to excel at a number of tasks which require understanding of abstract rules that govern the relationships between objects in a matter that surpasses the purely perceptual qualities of those objects (see e.g. (Jones & Love, 2007)).

Ongoing work in neuroscience aims to understand how our brains represent such abstract relationships. Early results identified cognitive maps (Tolman, 1948) and so-called relational memories (Cohen & Eichenbaum, 1993) which seem to encode abstract relational information invariant to some perceptual features (see also (Sedda & Scarpina, 2012; Goodale & Milner, 1992)). While this abstraction of relational knowledge from memory is not entirely disconnected from sensory modality (Barsalou et al., 2003), there is evidence that isolated perceptual features alone are not principally driving relational reasoning, and that complex interdependence of processed features and memory gives rise to abstraction (Goldstone et al., 1989).

Consistent with this neuro-cognitive picture, AI systems which work well on a variety of domains like machine translation (Vaswani et al., 2017; Dehghani et al., 2018; Devlin et al., 2018), image classification (Dosovitskiy et al., 2020), etc. do not perform as well on tasks that explicitly require inference of relational structure between entities (Webb et al., 2021; Johnson et al., 2017; Yi et al., 2019). An example of such a task is Raven's Progressive Matrices (Raven & Court, 1938) which, given a sequence of objects, tests the ability to infer the relations between objects and to use it for prediction of possible objects that satisfy the underlying relational structure. In response to this shortcoming, a number of AI elements inspired from neuroscience have been proposed to tackle relational abstraction (Santoro et al., 2018; Graves et al., 2014; Mittal et al., 2020; Pritzel et al., 2017; Hill et al., 2020; Fortunato et al., 2019; Webb et al., 2021; Whittington et al., 2019).

---

*Correspondence to Giancarlo Kerg: giancarlo.kerg@gmail.com
[1]Université de Montréal, [2]McGill University, [3]CIFAR Senior Fellow, [4]CIFAR AI Chair,
Code is available at: https://github.com/giancarlok/relationaltasks

A working hypothesis we explore here is that most artificial systems do not work well on relational reasoning because they do not encode an explicit notion of relations between different objects, and rather rely too much on representations of object features. In contrast, in our brains, relational dependencies across high-level entities form a key ingredient to our understanding of the world and augment our ability to reason in potentially unseen scenarios (Whittington et al., 2019), and this has been reflected in a large body of machine learning approaches exploiting relational structure (Scarselli et al., 2008; Veličković et al., 2017; Kipf et al., 2018; Battaglia et al., 2018; Bengio et al., 2019; Webb et al., 2020; Zhang et al., 2019).

However, recent work aims to address this shortcoming by explicitly focusing on encoding relational information between objects. Emergent Symbols through Binding in Memory (ESBN) (Webb et al., 2021) leverages the abstract nature of relations by explicitly preventing sensory information from informing relational encoding. While showcasing the importance of separate sensory and relational encoding, ESBN also includes a number of architectural design choices that may or may not be important for out-of-distribution generalization in purely relational tasks.

In this work, we explore the inductive bias of separating sensory information from abstract, relational representations. In doing so, we distill a minimal set of inductive biases which allow for superior generalization than ESBN. Through a series of numerical experiments involving purely relational tasks on data with scaled input size and variable context, we demonstrate and analyze why the simple act of building similarity relations between objects irrespective of sensory encoding is enough to improve a range of different model architectures. Thus, our results confirm the importance of abstract relational encoding and establish minimal inductive biases to enhance other system architectures. We find that focusing on these inductive biases to design new and simpler architectures considerably improves out of distribution generalization across all settings tested, and thus is a promising approach for other machine learning researchers to adopt.

## 2 BASIC RELATIONAL TASKS

We first describe the basic set of relational tasks to analyze inductive biases required for good OoD generalization: same/different, relational match-to-sample (RMTS), distribution-of-three and identity rules (Webb et al., 2021). In Section 5, we define a suite of harder tasks to study if these inductive biases extrapolate well.

In each of the tasks, a sequence of objects is provided which are related through a specific abstract rule. The models are tested for out-of-distribution (OoD) generalization by preserving the abstract rule but using novel objects unseen during training. We briefly recall the basic tasks:

*Same/different.* Two objects are shown, and the task is to determine if they are same or different (Figure 1a).

*Relational match-to-sample (RMTS).* Three pairs of objects are presented: a *source* and two *target* pairs. The task is to identify which target pair matches the source pair, i.e., if the source pair has two identical/different objects, the task is to identify the target pair with two identical/different objects (Figure 1b).

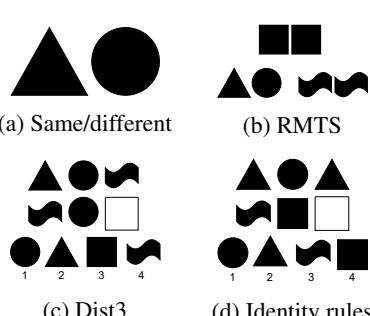

(a) Same/different (b) RMTS

(c) Dist3 (d) Identity rules

Figure 1: Basic relational tasks. (a) Same/different (answer: different). (b) Relational match-to-sample (answer: second target pair). (c) Distribution-of-three (answer: 2). (d) Identity rules (answer: 3).

*Distribution-of-three (Dist3).* Nine objects are presented, pictured as three rows (Carpenter, 1990). The second row is a permuted version of the first row, but with the last object hidden. The task is to identify the hidden object from the set of four choices shown in the third row (Figure 1c).

*Identity rules.* Nine objects are presented, pictured as three rows (G.F. Marcus, 1999). The first and second row consist of three objects governed by the same pattern (e.g. ABA or ABB or AAA) with the last object of the second row hidden. The task is to identify the hidden object from the set of four choices shown in the third row, by inferring the underlying pattern (Figure 1d).

While these problems appear simple, deep neural network architectures (including LSTM (Hochreiter & Schmidhuber, 1997), Transformers (Vaswani et al., 2017), Relation Networks (Santoro et al., 2017),

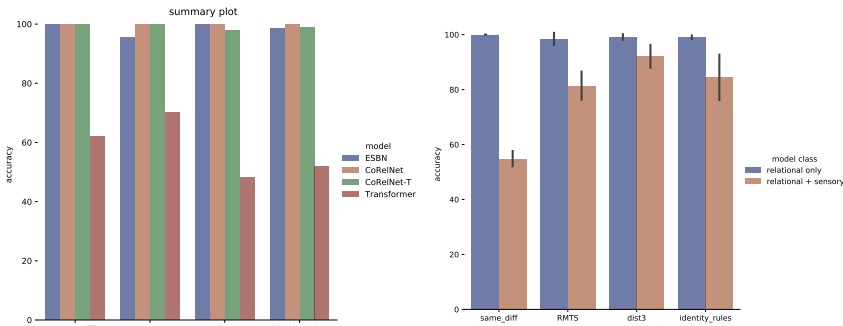

Figure 2: Test accuracy on the four basic relational tasks for (left) different models, and (right) relational only models (ESBN and CoRelNet) vs relational + sensory input models (here we concatenate the encoded sensory input vectors in both ESBN and CoRelNet). Results are averaged over 10 seeds and displayed for the extreme OOD setting. See Figure 6 and Figure 7 in the Appendix for details.

Predi-Net (Shanahan et al., 2019), NTM (Graves et al., 2014), MNM (Munkhdalai et al., 2019)) fail to generalize well in OoD settings in such relational tasks (Kim J, 2018; Webb et al., 2021).

## 3 MODELS

All models introduced in this section are directly inspired by ESBN (Emergent Symbol Binding Network) introduced in (Webb et al., 2021), which constructs an external memory through a recurrent controller and makes use of the inductive bias of separating sensory information from relations, and hence factoring the model into two distinct information processes.

Similar to the ESBN architecture, all the models introduced in this paper use an encoder $q$ applied to each of the inputs $\{x_t\}_{t=1}^T$ followed by a temporal context normalization (TCN) (Webb et al., 2020) step, which has shown to improve out-distribution generalization in relational reasoning tasks. We then apply self-attention onto the vectors $z_t = \text{TCN}(\{q(x_t)\})$, yielding the matrix $R \in \mathbb{R}^{T \times T}$ which encodes the relational information between the object representations $z_t$. Note that self-attention is computed via dot products and hence can be interpreted as giving rise to a similarity measure.

In our main model, *CoRelNet*, we simply flatten the matrix $R$ and feed it *directly into an MLP decoder*. We acknowledge that this architectural choice requires a hard-coding of the sequence length $T$ as the input dimension to the MLP decoder is $T^2$. In order to avoid hard-coding the sequence length $T$ explicitly, we also define a variation of *CoRelNet*, where the MLP decoder is replaced by a transformer, which we denote by *CoRelNet-T*.

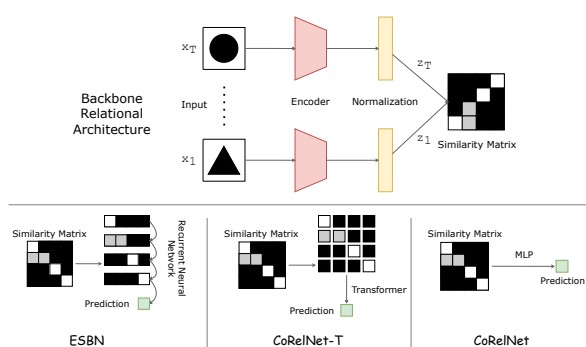

Figure 3: **Illustration of Relational Architecture.** (top) The backbone relational architecture describes the common backbone present in all of the relational architectures considered in this work, with a similarity matrix with entries for each pair of input objects. (bottom) ESBN considers a Recurrent Network to process the similarity matrix obtained at the end of the backbone while CoRelNet-T and CoRelNet use a Transformer and an MLP respectively.

We refer the readers to Figure 3, which describes the fundamental differences between the above models. In the backbone of all the relational architectures, each object $x_t$ is independently encoded into its representation $z_t$, which is then used to construct a similarity matrix. The fundamental difference between ESBN, *CoRelNet* and *CoRelNet-T* is the way this similarity matrix is processed to provide a downstream prediction. In particular, ESBN employs a recurrent neural network with

a writing procedure into external memory, while *CoRelNet-T* utilizes a transformer network and *CoRelNet* utilizes just a simple MLP system to operate on the matrix directly and provide a prediction.

We will see in later experiments (section 5) that directly accessing the similarity matrix *all at once*, as opposed to passing the relational information sequentially through the bottleneck of the hidden state of a recurrent controller, will provide an advantage in more complex settings involving not only unseen objects, but also unseen relations underlying those objects.

## 4  ALL YOU NEED IS THE SET OF SIMILARITY SCORES

Motivated by the results presented in ESBN, we want to evaluate the hypothesis that *the relational information between the images seen in the input sequence is all we need for OoD generalization on these relational tasks.* Hence, looking for a minimal set of inductive biases, we first investigate how important for OoD generalization is the said disconnection from the sensory information.

We first note that given the structure of the basic tasks, the ground-truth prediction rule is defined as a function of the relationships between the objects of the scene and is completely de-coupled from the actual shapes of individual objects. In particular, the rules do not take into account the identity of the objects but instead only rely on which objects are the same and which aren't. Since this is the basis for the ground-truth prediction rule, we hypothesize that if a model learns this function and ignores any information regarding the absolute identity of the objects, then it will be able to generalize well in the OoD settings where the only change performed is the identity of objects considered.

Our experiments indeed show that the models that make predictions solely on top of the similarity scores between objects (*ESBN*, *CoRelNet* and *CoRelNet-T*) do exceptionally well on OoD generalization, as opposed to models that also rely on the sensory information about what the objects are (*Transformer*, *LSTM*, *RN*, etc). This is illustrated in Figure 2 for our four tasks as well as in the results outlined in ESBN. This clearly makes the point that just the relational information in the form of a $T \times T$ similarity matrix between objects is sufficient to nail these tasks, where $T$ is the number of objects in the sequence.

Next, we also show experimentally that a complex architecture like *ESBN* or *CoRelNet-T* on top of the similarity scores is not needed and that a simple MLP, like in *CoRelNet*, actually outperforms the other models on OoD generalization (also see Figure 5). Thus, having stripped away all the additional inductive biases regarding writing keys and recurrence from *ESBN*, we actually see better performance and optimal OoD generalization.

As an additional evidence for the hypothesis of this section, we see in Figure 2 that the concatenation of sensory information to the already present relational information in input to the classifier stage degrades the OoD generalization capacity of the models. This is because overfitting on the sensory information of objects cannot lead to generalization in the OoD settings, while memorizing the similarity matrix patterns does generalize on unseen new shapes, since the prediction rule as a function of the similarity matrix remains the same in as well as out of distribution.

## 5  DO THESE INDUCTIVE BIASES EXTRAPOLATE?

In this section, we study how the mentioned inductive biases extrapolate to harder tasks. We construct harder versions of the basic relational tasks by restricting not only the set of objects but also the set of relations seen during training, while testing on unseen objects as well as unseen relations. We first describe the set of tasks that test the model's capacity for generalizing to both unseen objects as well as relations.

*RMTS 3.* Similar to RMTS, we have a *source* triplet and two *target* triplets. The task consists of identifying which target triplet has the same relational structure as the source triplet. A triplet can either consist of (a) a repetition of the same object ($AAA$), or (b) two distinct objects ($ABA$,

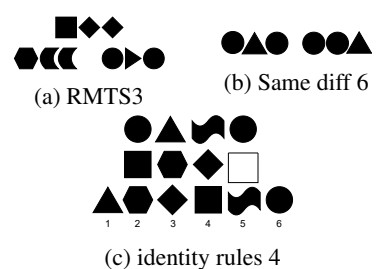

(a) RMTS3          (b) Same diff 6

(c) identity rules 4

Figure 4: Harder relational tasks. (a) Relational match-to-sample 3 (answer: first target pair) (b) Same/different 6 (answer: different). (c) Identity rules 4 (answer: 4).

$BAA$ or $AAB$), or (c) three distinct objects ($ABC$). For training, only triplets of the form ($AAA$, $ABA$ and $BAA$) are shown, while testing *only* involves triplets of the form ($ABC$, $AAB$). Hence, we test on both unseen objects and *unseen relations*.

*Same/diff 6.* The task consists of 6 objects (two triplets - first 3 objects and the last 3 objects). The goal is to determine whether the two triplets are identical or not. Here, each triplet consists of at most two distinct objects (thereby of forms $AAA$, $AAB$, $ABA$ or $BAA$). During training, only triplets of the form ($AAA$, $ABA$, $BAA$) are shown, while evaluation is done on examples that have at least one triplet of the form ($AAB$).

*Identity rules for 4 elements.* Similar to the original identity rules task, we consider three rows of object as shown in Figure 4c. The task consists of predicting the fourth element in the second row (from the set of options in the last row) such that the relational structure is preserved between the first two rows. During training, only quadruples with at most two distinct objects are shown, while testing relies *only* on quadruples with exactly three distinct objects. Similar to *RMTS 3*, we are not only testing on unseen objects, but also on unseen relations.

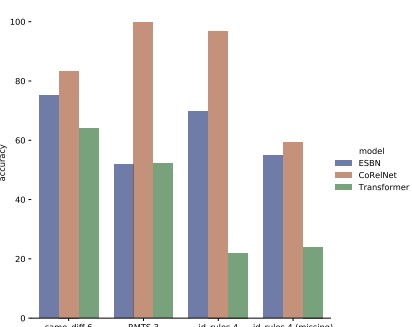

*Identity rules for 4 elements (with missing variations).* This task is similar to *identity rules for 4 elements* described above. The only difference is that the training set does not include quadruples of the form ($ABAA$, $ABAB$).

**Results.** We see that *CoRelNet* outperforms ESBN and Transformer on all four tasks (Figure 5), with near perfect accuracy on *RMTS 3* and *identity rules 4*, confirming that it is not only able to generalize to unseen objects in more complex settings, but also to unseen relations. We hypothesize that the drop in performance of all models in the *identity rules for 4 elements (with missing variations)* task is due to the fact that the "essential base elements" that span the space of all possible abstract relations are left out. We specifically set up this task in order to see a substantial drop in performance. We also note that leaving out either $ABAA$ or $ABAB$ is not sufficient for *CoRelNet*

Figure 5: Performances on the 4 harder tasks with unseen relations. Results are displayed for the most extreme OOD case and reflect test accuracies averaged over 10 random seeds. For full details see Figure 8 in the appendix.

to experience a drop in performance, both classes of quadruples need to be removed from the training set.

## 6 CONCLUSION

We conclude that for purely relational tasks, the relational information between the objects is all that is needed for OoD generalization. In particular, disconnecting the said relational information from the sensory input is an essential inductive bias for OoD generalization on unseen inputs. This is further strengthened by the fact that after removing additional inductive biases in the decoder, we still achieve good OoD generalization as long as prediction is driven *solely* by relational information. In other words, OoD performance on purely relational tasks without spurious inputs is a direct reflection of how well the decoder translates the relational information contained in the similarity matrix to the correct output. It turns out that a simple MLP decoder does the job, accomplishing good OoD generalization not only on unseen objects but also on unseen relations. We also note that OoD performance for unseen inputs and unseen relations can deteriorate if certain crucial classes of training examples are withheld. Finally let us emphasize that the MLP requires a hardcoding of the sequence length.

Future work includes understanding how to deal with tasks where multiple abstract rules are at play and need to be inferred. Secondly, one could explore how to improve performance in more real world settings involving a variety of spurious features. Finally, one could investigate key inductive biases that enable artificial systems to generalize to more complex unseen relations from a minimal set of training examples.

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

## ACKNOWLEDGEMENTS

GK acknowledges UNIQUE PhD Excellence Scholarship. SM acknowledges UNIQUE Masters Excellence Scholarship. The authors are grateful to Samsung Electronics Co., Ldt., CIFAR, and IVADO for their funding and Calcul Québec and Compute Canada for providing us with the computing resources.

## APPENDIX

## A DETAILS OF IMPLEMENTATION

### A.1 ENCODER

All inputs are $32 \times 32$ grayscale images containing a single Unicode character, except for the spurious tasks, where the input is of shape $3 \times 32 \times 32$. Each input is fed sequentially and separately to the encoder. The encoder consists of 3 convolutional layers with 32 channels, a $4 \times 4$ kernel and a stride of 2, followed by two fully connected layers with 256 and 128 hidden units respectively. All layers have ReLU nonlinearities. All weights were initialized using a Kaiming normal distribution (He et al., 2015) and all biases were initialized to 0.

### A.2 SIMILARITY MATRIX

The output dimension of the encoder is 128, and hence after applying temporal context normalization (TCN, Webb et al. (2020)) the input dimension to the similarity matrix is 128. The output of the similarity matrix is $T \times T$, where $T$ is the sequence length.

### A.3 DECODER

For the CoRelNet model, only the lower triangular part of the matrix is being used and flattened into a vector of size $T \cdot (T + 1)$ which is fed to a simple MLP with one hidden layer of 256 units and ReLU nonlinearity. For CoRelNet-T, again only the lower triangular part of the matrix is being used and the (strict) upper triangular matrix is zeroed out. One then uses a multi-head attention architecture similar to the one in section 2.1 in (Mittal et al., 2022), with 8 heads, 1 layer, and dimension 512.

### A.4 TRAINING

All models were trained with a batch size of 32 using the ADAM optimizer (Kingma & Ba, 2014) and a learning rate of $5e - 4$, except on the *same/different* 6 task the batch size was $12 \cdot 32 = 384$, on the *RMTS* 3 task the batch size was $6 \cdot 32 = 192$, on the *identity rules 4* task the batch size as $7 \cdot 32 = 224$, and on the *identity rules 4 with missing examples* task the batch size was $5 \cdot 32 = 160$.

# B  BASIC RELATIONAL TASKS AND MODELS

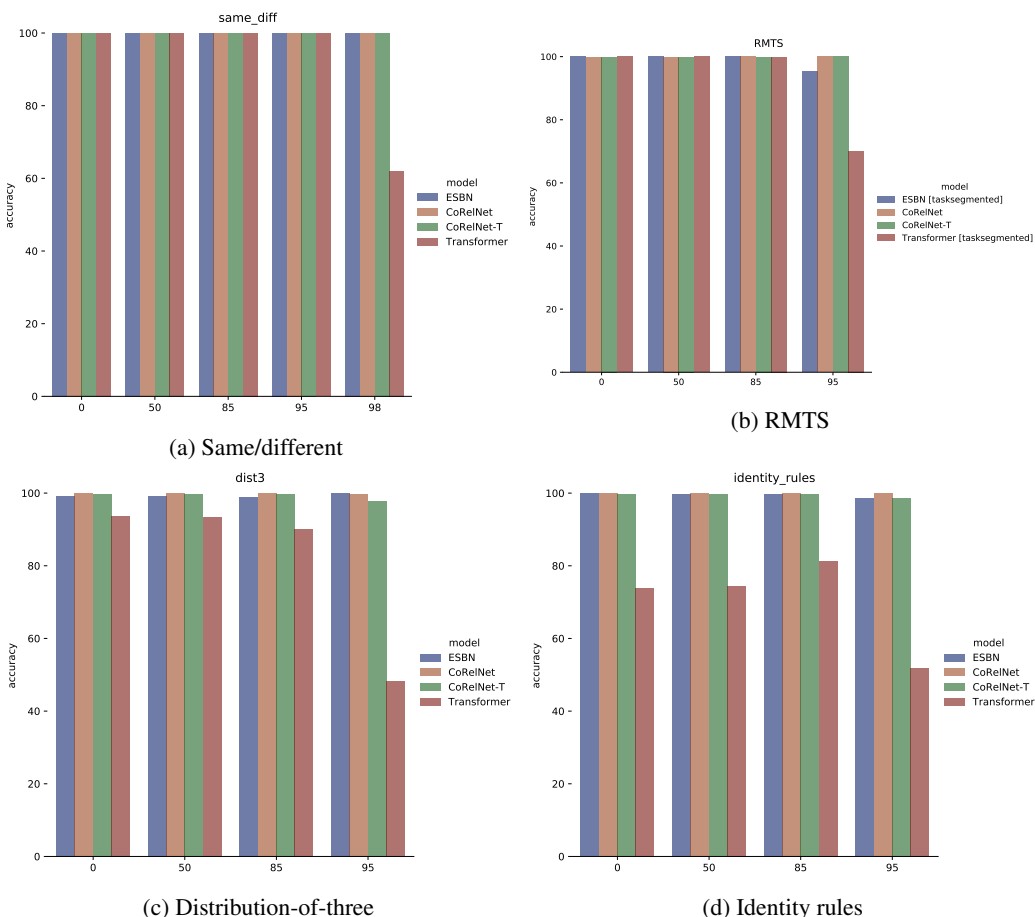

Figure 6: Full detailed test accuracy results for the four basic relational tasks, across the full range of values for $m$ (the number of heldout shapes during training only shown at testing, displayed on the $x$-axis). There are total of $n = 100$ shapes, hence $100 - m$ of those are shown during training, and the test set consists only of the other $m$ shapes. The most extreme OoD case corresponds to $m = 98$ for the same different task (98% of the possible combinations of shapes are in the test set and not shown during training), and $m = 95$ for the 3 other tasks. The case $m = 0$ corresponds to the in-distribution case, where the same 100 shapes are shown at testing and training. The test result accuracies are averaged over 10 random seeds.

# C  BASIC RELATIONAL TASKS WITH SENSORY INFORMATION ABLATION

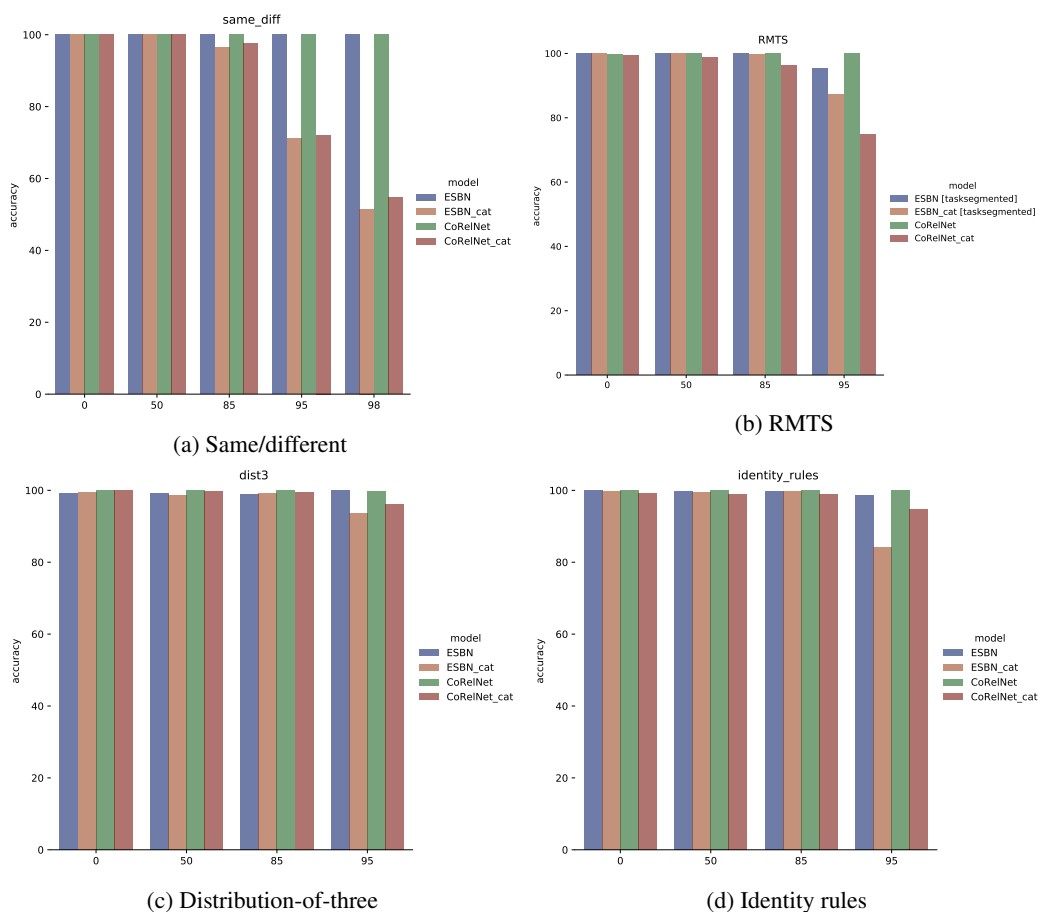

Figure 7: Full concatenation plots. Full detailed test accuracy results on the four basic relational tasks, across the full range of values for $m$ (the number of heldout shapes during training, displayed on the $x$-axis). There are total of $n = 100$ shapes, hence $100 - m$ of those are shown during training, and the test set consists only of the other $m$ shapes. The case $m = 0$ corresponds to the in-distribution case, where the same 100 shapes are shown at testing and training. Here the suffix "cat" stands for concatenating the encoded input on top of the input of the decoder. The test result accuracies are averaged over 10 random seeds.

## D  HARDER TASKS WITH UNSEEN RELATIONS

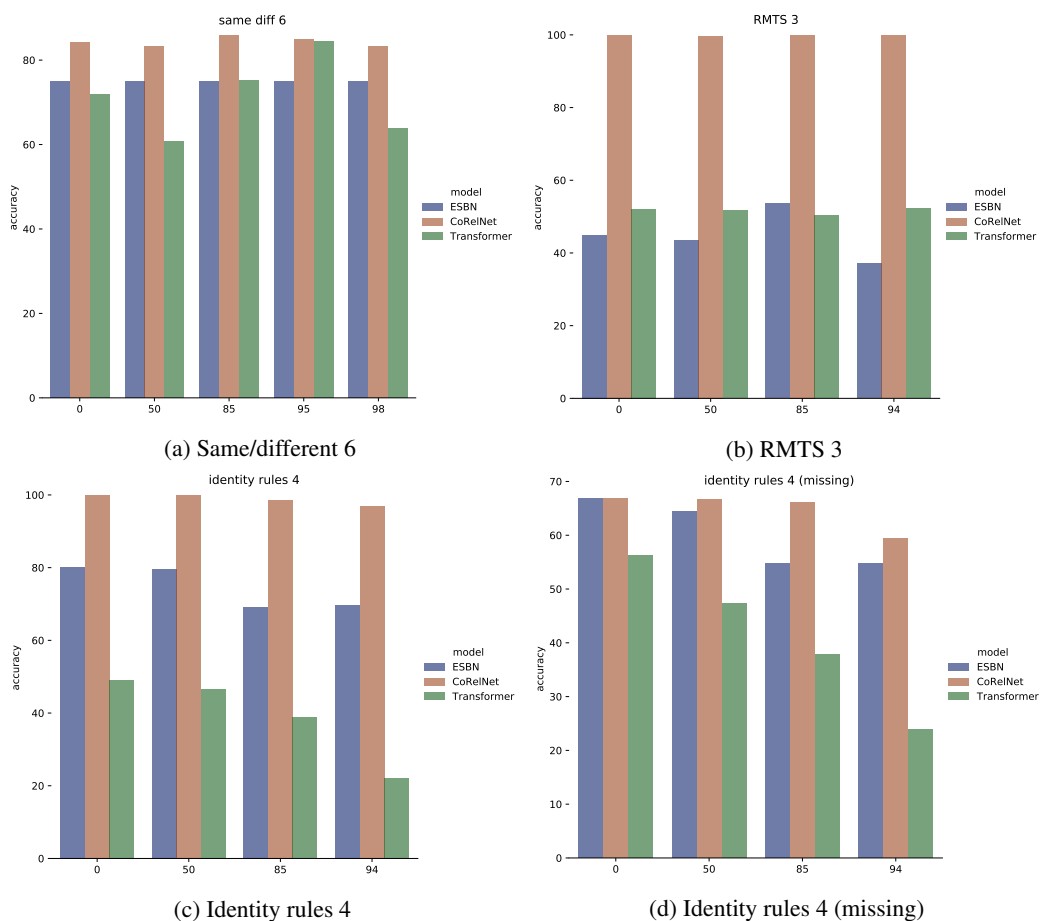

Figure 8: Full detailed test accuracy results on the harder relational tasks with unseen relations, across the full range of values for $m$ (the number of heldout shapes during training, displayed on the $x$-axis). There are total of $n = 100$ shapes, hence $100 - m$ of those are shown during training, and the test set consists only of the other $m$ shapes. The case $m = 98$ corresponds to the most extreme OoD case for same/different 6, and $m = 94$ for the other three tasks. The case $m = 0$ corresponds to the in-distribution case, where the same 100 shapes are shown at testing and training. The test result accuracies are averaged over 10 random seeds.

