# OpenReview forum: "Inductive Biases for Relational Tasks"
_ICLR.cc/2022/Workshop/OSC — ICLR2022 OSC  Poster_

### Official Review · Reviewer_6ypX · 2022-03-12
**INDUCTIVE BIASES FOR RELATIONAL TASKS**

**Rating:** 2
**Confidence:** 3

**Review:**

Summary
This paper considers several relational tasks over simple, 2D shapes. The authors make a neuroscience argument for separating sensory inputs from relational reasoning, and extend an existing model, demonstrating better results on several tasks.

Pros
-Strong neuroscience background
-Interesting insight, different from prevailing GNN methods of relational reasoning

Cons
-Not really clear how inductive bias of separating perception from relations is implemented--CoRelNet still derives its similarity matrix from percepts
-Hard to understand CoRelNet well without understanding seperate paper, ESBN. Consider including more network description in paper.
-Would be nice to see wider varity of tasks/visual inputs. Consider adding tasks from https://arxiv.org/abs/1911.01547

Other feedback
-Figure 2, plots missing (a) and (b) labels
-Please add error bars and confidence intervals to all results

---

### Official Review · Reviewer_smGX · 2022-03-15
**Restricted applicability, but somewhat interesting results**

**Rating:** 2
**Confidence:** 3

**Review:**

* This paper explores a variety of simple relational tasks and how different relational inductive biases play a role in generalization on these tasks. The experiments seem fairly principled, and the overall questions are of some interest. However, the significance of the work is limited by the fact that the tasks are strictly relational—that is, none of the item-level information (what the authors refer to as "sensory details") is ever relevant to the task, only the relationships. This is an extremely restricted setting that will obviously lead to different conclusions than cases when both relational and feature level information is relevant, e.g. giving a system a task like "Get the object that's the same shape as the red object, but a different color." Our brains are clearly capable of representing both types of information. Of course, the experimental settings here are more controlled because of this restriction to purely relational tasks, but the abstract/introduction/conclusions could be more specific about emphasizing this aspect of the setting (including both its benefits and limitations).

* For the transformer, did the authors treat the flattened similarity matrix as a sequence? Using a transformer affords the possibility of respecting the 2D structure by using 2D position embeddings, as are used in vision transformers (or some other set of embeddings which respects the row and column structure of the matrix. This might improve the transformer's performance.

* More broadly, details of the implementation could potentially make a difference here, so it would be useful to specify the details of each architecture and the training process in the appendix (e.g. image size, how many layers in the encoder and the MLP, embedding dimension over which the similarities are computed, etc. etc.)

* As the authors note, the MLP decoder the authors propose is restricted to similarity matrices of the same size, while the other approaches like ESBN are not if I understand correctly; this seems to merit slightly more discussion in the intro + conclusions when describing the contributions of this work.

---

### Decision · Program_Chairs · 2022-03-24

Accept (Poster)